

# Could hand-eye laterality profiles affect sport performance? A systematic review

Miquel Moreno[1,2], Lluis Capdevila[1,3] and Josep-Maria Losilla[3,4]

[1] Laboratory of Sport Psychology, Department of Basic Psychology, Universitat Autònoma de Barcelona, Bellaterra, Catalunya, Spain
[2] Department of Sport Sciences, Universitat de Vic, Vic, Barcelona, Catalonia, Spain
[3] Sport Research Institute, Universitat Autònoma de Barcelona, Bellaterra, Catalunya, Spain
[4] Departament of Psychobiology and Methodology of Health Science, Universitat Autònoma de Barcelona, Bellaterra, Catalunya, Spain

Corresponding author
Lluis Capdevila,
lluis.capdevila@uab.cat

## ABSTRACT

**Background:** Laterality effects on sports performance have been a field of interest for the sports sciences, especially in asymmetrical sports, which require the preferential use of one side of the body. Some sports in particular involve the visual system and ocular laterality, due to the need to clearly focus on a dynamic object (ball, opponent, projectile, *etc.*). The relationship between manual and ocular laterality results in two perceptual-motor profiles, one where the dominant hand and eye are ipsilateral (uncrossed hand-eye laterality profile, UC-HELP), and the other where they are contralateral (crossed hand-eye laterality profile, C-HELP).

**Methodology:** A systematic review of the literature was carried out to determine the prevalence of hand-eye laterality profiles in the different sports modalities and their relationship with psychological factors and sports performance. Searches of PsycInfo, Medline, Scopus and grey literature identified 14 studies (2,759 participants) regarding hand-eye laterality in sports that met the eligibility criteria.

**Results:** Previous studies have estimated that between 10–30% of the general population exhibit a C-HELP, and 70–90% have an UC-HELP. The results of the reviewed studies indicate that in some sports the percentage of C-HELP is higher in regular and high-level athletes than in the normal population: golf (52.55%), soccer (53%), tennis (42%) and team sports (50.7%). In target sports (archery and shooting) athletes with an UC-HELP seem to have an advantage given the significant concentration of this profile in the highest performing populations (82.3%). In basketball, cricket and golf, the literature reviewed also reported biomechanical differences in the execution of some techniques between the two profiles. We did not find any study in our review that related hand-eye laterality with cognitive, tactical, or psychological aspects of athletes.

**Conclusions:** These results should be taken with great caution due to the potential bias linked to the methodologies used in the investigations, the heterogeneity in the assessment of hand-eye laterality, the few studies available on the subject and the indirect nature of many of the observed relationships between performance and laterality. For further investigation, we propose a standardized terminology and protocol of hand-eye laterality assessment in sports. The advancement in knowledge about hand-eye laterality profiles, along with the study of the relationship with psychological or tactical-sports patterns, can contribute to more effective development plans for athletes and can be a complement to talent detection.

# INTRODUCTION

Laterality is the preferential use of one part of the body with respect to its symmetrical side. This phenomenon has been a subject of scientific interest and it's been researched in fields like biology and psychology (*e.g.*, *MacManus, 2002*; *Rogers, Vallortigara & Andrew, 2013*). The relationship between two types of laterality is examined here: handedness, commonly defined as the preference of one hand over the other in unimanual tasks (*Scharoun & Bryden, 2014*); and eyedness or eye-dominance, the preference for visual input from one eye over the other. The dominant eye provides more input to the visual cortex and relays information more accurately, such as the location of objects, and it is observed when monocular images cannot be fused or when monocular viewing is required (*Valle-Inclán et al., 2008*). The first publication regarding the relationship between handedness and eyedness dates back to the 16th century, when *Porta (1593)* defined hand-eye laterality profiles and introduced the first eyedness measurement test. This relationship is significant for activities that require coordination of the eyes (as receptor organs) and the limbs (as effector organs) for accurate response. In this kind of task, manual responses are lateralized in the contralateral hemisphere while the dominant eye is functionally connected to the ipsilateral hemisphere (*Azémar, Stein & Ripoll, 2008*). There are two types of hand-eye laterality profiles: one results from having the same side of preference for both hand and eye (uncrossed hand-eye laterality profile, UC-HELP), and the other from having eye and hand preference on different sides of the body (crossed hand-eye laterality profile, C-HELP).

Ever since *Orton (1925)* pointed out a relationship between C-HELPs and reading difficulties in children, crossed laterality has received considerable study in the field of literacy which supports the association between C-HELPs and neurological problems that may result in poor reading performance (*e.g.*, *Orton, 1937*; *Vernon, 1971*; *Kershner, 1975*; *Abigail & Johnson, 1976*; *Richardson & Firlej, 1979*). Some studies have linked C-HELPs with specific cognitive disorders. For example, *Porac & Coren (1976)* found that the C-HELP was more prevalent in individuals manifesting a variety of behavioral disorders, and *Nagae (1983)* showed that C-HELP children performed significantly worse at verbal self-regulation of motor behavior, supporting the view that the functions of cerebral hemispheres in C-HELP children were more immature and linked with learning disabilities. However, a meta-analysis by *Bourassa, MacManus & Bryden (1996)* with 54,087 participants from 47 studies on hand-eye laterality did not find enough evidence to associate hand-eye laterality with learning and indicated the necessity of conducting more research in the field. In a more recent systematic review, *Ferrero, West & Vadillo (2017)* also found a lack of scientific evidence on the relationship between C-HELPs, academic achievement, and intelligence.

Determining the prevalence of C-HELPs in the general population has also been the subject of various studies. *Robinson, Jacobsen & Heintz (1997)* compiled a multi-site

sample of 1,005 participants and reported a C-HELP prevalence of 41.4%. The above cited meta-analysis by *Bourassa, MacManus & Bryden (1996)* found a 34.8% prevalence of C-HELPs. In another meta-analysis with 10,635 participants from 14 studies, *MacManus et al. (1999)* used the throwing hand and the writing hand as criteria to assess handedness and observed a C-HELP prevalence of 25.4% with respect to the throwing hand and of 25.8% with respect to the writing hand.

Sports that are considered asymmetric have been more deeply studied since they imply the preferential use of one of the two sides of the body to throw, hit or use implements. These include tennis (*Dallas, Mavvidis & Ziagkas, 2018*), golf (*Dalton, Guillon & Naroo, 2015*; *Sugiyama & Lee, 2005*), baseball (*Laby et al., 1998*; *Classe et al., 1996*; *Portal & Romano, 1988*), cricket (*Thomas, Harden & Rogers, 2005*) and basketball (*Shick, 1971*, *1977*; *Lopez-Diaz et al., 2015*). Several studies have analyzed the relation between the distribution of laterality profiles and their effects on sports performance. *Azemar (2003)*, in a survey of 1,707 participants (including 229 normal controls, 1,126 sports students and 352 elite athletes), observed that the prevalence of C-HELP was significantly higher in tennis, fencing, boxing and gymnastics, and significantly lower in archery, when compared to normal population values. These authors also pointed to a significantly higher percentage of C-HELPs in duel or adversary sports (47.8% in tennis, fencing and boxing) compared with non-adversary (35% C-HELPs in gymnastics and archery). Significant differences between sports modalities have also been reported in a study from *Quevedo et al. (2014)* with a sample of 536 elite multi-sport athletes, where a C-HELP prevalence of 55% (95% CI [44.03–65.97%]) was observed in golf, compared to a prevalence of 9% (95% CI [2.69–15.31%]) in shooting. Some authors have hypothesized about specific physiological advantages for the performance of certain tasks in C-HELP subjects. For example, *Azemar & Ripoll (1987)* observed a visuo-motor advantage in response time for C-HELP subjects compared to UC-HELPs in laboratory experiments with spatio-temporal tasks. *Dorochenko (2009)* also raised the possibility of the existence of differences in personality and mental performance to explain a hypothetical over-representation of C-HELPs in the sport of tennis. In this same sense, *Laborde et al. (2009)* reported that knowledge of hand-eye laterality could be reliably used to advise sports training to enable more efficient adaptations in talent detection, learning skills and in achieving better levels of coordination. Nevertheless, *Laby & Kirschen (2011)* have warned about the lack of consensus among researchers on whether C-HELPs or UC-HELPs could be advantageous in various sports.

More research is needed to determine the practical applications of hand-eye laterality in training and to clarify the differences in hand-eye laterality profiles reported so far between sports modalities. The present systematic review aims to analyze the literature available to date on hand-eye laterality profiles in the different sports modalities, with three specific objectives: (a) to estimate the prevalence of hand-eye laterality profiles, (b) to examine the relationship between hand-eye laterality profiles, psychological factors and sports performance, and (c) to propose a methodological and terminological consensus.
## METHODS

The protocol for this systematic review was registered with the International Platform of Registered Systematic Review and Meta-Analysis Protocols (INPLASY) on 28 November 2020 (registration number INPLASY2020110127; DOI 10.37766/inplasy2020.11.0127). The study was undertaken in accordance with the Preferred Reporting Items for Systematic Reviews and Meta-Analyses (PRISMA 2020) statement (*Page et al., 2021a*, *2021b*). The Ethics Commission for Human Experimentation of The Universitat Autònoma de Barcelona granted Ethical approval to carry out the study (protocol code CEEAH-5745).

### Search strategy

Literature searches were performed using the following databases: PsycInfo by EBSCOhost, Scopus by Elsevier, Medline by PubMed, and Dissertations & Theses Global by ProQuest. To include grey literature, we also searched in Google and reviewed up to 100 links. In addition, search alerts in PsycINFO and Scopus were set until December 2020.

The search strategy followed the recommendations of the Peer Review of Electronic Search Strategies (PRESS) guidelines (*McGowan et al., 2016*). With the aim of identifying studies about hand-eye laterality and sports and due to the lack of consensus in the use of the terms for this domain of knowledge, the search strategy included a long string of synonyms and related terms. The search was limited by population (humans), by language (English, French or Spanish) and by publication type (peer reviewed journals). The specific search syntax used for each database can be found in Appendix 1.

### Eligibility criteria and study selection

Eligible studies had to fulfill the criteria of being original empirical studies (experimental, quasi-experimental, observational, or single-case designs) providing direct information on hand-eye laterality (distribution, predictiveness and influence on sports performance, or any correlation with psychological factors).

No exclusion criteria were applied by gender, age, or temporal limit of the publication. Although there is currently great interest in hand-eye coordination in electronic games, our focus was on traditional sports, so studies referring to e-sports, virtual reality or gaming were excluded from our review.

One reviewer (MM) applied the inclusion/exclusion criteria to all titles and abstracts. Studies meeting the eligibility criteria were selected and studies that could cause controversy regarding the inclusion/exclusion criteria were also pre-selected and the full text was retrieved as well. The pre-selected papers were checked independently by two review authors (MM, LC). Discrepancies were resolved through discussion with a third author (JML) where necessary until reaching consensus.

### Data extraction

A data extraction template was previously designed to extract data from the included studies. Extracted information included: study characteristics (authors, title, year, journal, research design); sample information (size, mean age, sex distribution, sports disciplines,

population/country, *etc.*), and hand-eye laterality data (handedness test, eyedness test, C-HELP and UC-HELP distribution by sports modalities and sex, effects of HEL on performance, skills analysed, relationships between HEL and psychological traits, *etc.*). Data extraction was carried out independently by two reviewers (MM, LC) and discrepancies were resolved through discussion with a third author (JML) where necessary.

## Strategy for data synthesis

This review provides a narrative and tabular synthesis of the data extracted from the included studies, structured around the research design, sport discipline and other factors of interest. The main information is shown in tables. In the discussion, some information about the findings of the review and how these findings may guide further research is reported.

## Risk of bias assessment

The critical appraisal checklist for analytical cross-sectional, prevalence and quasi experimental studies proposed by the Joanna Briggs Institute (JBI) (*Moola et al., 2020*) was applied to assess the risk of bias of the selected studies (Appendix 2). No studies will be excluded due to high risk of bias because the amount of risk of bias is a relevant result in and of itself in our review.

The risk of bias was evaluated independently by two review authors (MM, JML). Discrepancies were resolved through discussion with a third author (LC) where necessary.

# RESULTS

## Literature search

Figure 1 shows the flow diagram for systematic reviews of scientific literature proposed by PRISMA. After duplicate records in the databases were excluded, a total of 1,297 potential studies regarding hand-eye laterality in sports were identified. There was 100% agreement during the selection phase without the need for the participation of the third reviewer. In the end, 14 studies were considered for this review for the qualitative synthesis of the data.

The demographic data extracted from the reviewed studies is shown in Table 1, and the main results found in the reviewed studies are shown in Table 2.

## Distribution of the age, gender, and geographical origin of the participants in the selected studies

A total number of 2,759 participants have been studied in the selected studies. Considering the distribution by age, we have a 2.5% of children (up to 12 years), 19.4% of teenagers (13–18 years), and 78.1% of adult population (older of 18 years). Only two studies (14.2%) were carried out with children and adolescents (9–17 years); five studies (35.7%) were carried out with college students, but not all of them reported the participants ages; five other studies (35.7%) selected samples of high performance athletes with ages ranged between 16 and 35 years old; five studies (35.7%) used amateur athletes or sports practitioners (16.9–31.3 years); and finally, three of those studies (21.4%) compared data

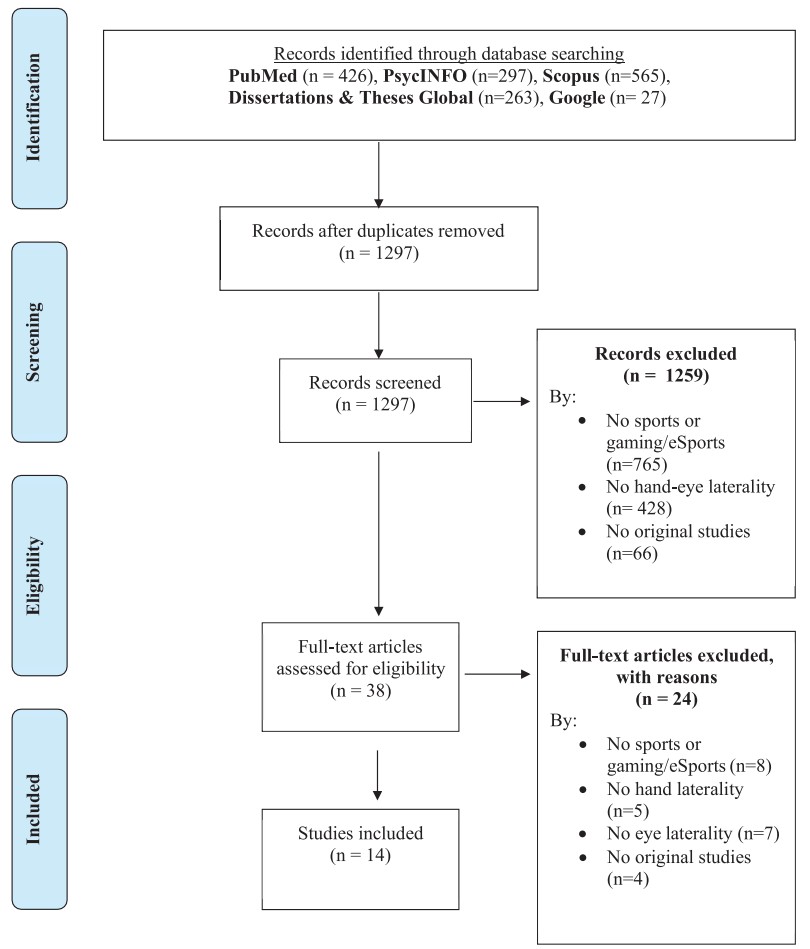

**Figure 1 PRISMA flow chart of the process of identifying and selecting studies.**

between professional (16–35.2 years) and amateur athletes (16.9–31.3 years). In the section on the terminology used, we detail the methodology used to determine the level of sports practice.

Geographical analysis of the selected studies revealed that eight of them (57%) were performed in Europe (including two in France, two in Spain, one in Greece, one in the Czech Republic and two in the United Kingdom), four (29%) studies were performed in the United States and two studies (14%) in Asia (Japan and Iran).

## Study publication dates

Study publication dates ranged between 1971 and 2020, skewed heavily towards the last two decades (Fig. 2), with more than half of the studies in this period (57% between 2010–2022).

## Risk of bias of the selected studies

The vast majority (78.5%) of studies have implemented cross-sectional designs, two studies (14.4%) used a quasi-experimental pre-post design without a control group, and one was a

**Table 1 General characteristics of the reviewed studies.**

| Study | Country | Sport | *n* (men) | Age ± Standard deviation | Research design |
|---|---|---|---|---|---|
| *Dalton, Guillon & Naroo (2015)* | United States | Golf | HPA: 10 (–)<br>RA: 7 (–)<br>BA: 14 (–) | – | CS |
| *Laborde et al. (2009)* | France | Archery | BA: 82 (48)<br>RA: 1,323 (–) | BA: 19.3 ± 1.7<br>RA: – | CS |
| *Lopez-Diaz et al. (2015)* | Spain | Basketball | RA: 34 (24) | 12.94 ± 0.35 | QE |
| *Mann, Runswick & Allen (2016)* | England | Cricket | HPA: 43 (43)<br>BA: 93 (93) | HPA: 29.6 ± 5.6<br>BA: 24.1 ± 7.2 | CS |
| *Nosek, Hurdálková & Cihlář (2018)* | Czech Republic | Biathlon | RA: 37 (–) | 16.4 ± 1.24 | CS |
| *Pointer (2008)* | United Kingdom | Motorsports | RA: 60 (54) | 19.9 ± 9.6 | CS |
| *Portal & Romano (1988)* | United States | Baseball | RA: 23 (–)<br>NA: 100 (–) | – | CS |
| *Quevedo et al. (2014)* | Spain | Multiple sports | RA: 536 (315) | 17.4 ± 3.7 | CS |
| *Razeghi (2012)* | Iran | Darts | BA: 20 (20) | 21.43 ± 1.33 | QE |
| *Shick (1971)* | United States | Basketball | RA: 32 (0) | – | CS |
| *Shick (1977)* | United States | Basketball | RA: 86 (0) | – | CS |
| *Sugiyama & Lee (2005)* | Japan | Golf | RA: 47 (37) | 20.2 ± 0.8 | CS |
| *Dallas, Mavvidis & Ziagkas (2018)* | Greece | Tennis | HPA:50 (50) | – | PR |
| *Zouhal et al. (2018)* | France | Soccer | HPA: 72 (72)<br>RA: 9 (9) | HPA:18.2 ± 2.2<br>RA: 19.6 ± 2.1 | CS |

**Note:**
HPA, high-performance athletes; RA, Regular athletes; BA, beginner athletes; NA, non-athletes; PR, prevalence; CS, cross-sectional; QE, quasi-experimental; –, not reported.

prevalence study (7.1%). None of the studies were implemented with an experimental design.

The application of the risk of bias assessment tools proposed by the JBI for the research designs of the selected studies shows a moderate or high presence of bias in most of them (Appendix 2). Nine of the cross-sectional studies do not clearly define the criteria for inclusion in the sample (Q1), and also nine of them don't identify or treat potential confounding factors (Q5, Q6). Half of these studies also do not measure the exposure (Q3) and the condition studied (Q4) in a valid and reliable way. The prevalence study that was included in the review fails four of the five risks of bias assessed, and the two quasi-experimental studies fail a third.

## Sports studied

Only one study (*Quevedo et al., 2014*) analyzed hand-eye laterality in a multisport perspective including acrobatics (gymnastics and synchro), combat (taekwondo, wrestling, and judo), team sports (soccer, volleyball, handball, basketball, hockey, softball, and water polo), skiing, motorsport, modern pentathlon, golf, shooting, swimming, athletics, weightlifting and racket sports (tennis and table tennis). Three studies (21.4%) were focused on basketball, two of the studies (14.3%) were focused on golf, and for the rest of

**Table 2 Main results on the relationship between hand-eye laterality and sports performance and skill level.**

| Study | Sport | HELP terminology | C-HELP%, UC-HELP% | Handedness assessment | Eye preference assessment | HELP and sports performance/ skill level relationship | Favourable direct effects | Favourable indirect effects |
|---|---|---|---|---|---|---|---|---|
| *Dalton, Guillon & Naroo (2015)* | Golf | Crossed, Uncrossed dominance | HPA (50, 50) RA (80, 20) BA (14.4, 76.6) | Author self-report questionnaire | Pointing Test | The distribution of C-HELP and UC-HELP was statistically different between the different skill groups | – | C-HELP |
| *Laborde et al. (2009)* | Archery | Crossed, Uncrossed laterality | BA (34.1, 65.9) RA (17.7, 82.2) | Edinburgh Inventory | Pointing Test | An analysis of variance indicated that beginners with an uncrossed pattern scored significantly more points than those with a crossed pattern | UC-HELP | – |
| *Lopez-Diaz et al. (2015)* | Basketball | Crossed, Homogeneous laterality | RA (27.8, 72.2) | Harris Test | Sighting Test | Over-representation of C-HELP at young high-level basketball players. Technical effect found: the shoot mechanics should be adapted on UC-HELP players | Biomechanical effects | |
| *Mann, Runswick & Allen (2016)* | Cricket | Do not refer to this relation | HPA (26, 74) BA (19, 71) | Edinburgh Inventory | Pointing Test | Technical effects found: placing the dominant hand in the top of the bat (reverse stance) offer a very significant advantage. Placing the dominant eye in front of the stance did not affect the performance | – | – |
| *Nosek, Hurdálková & Cihlář (2018)* | Biathlon | Crossed, Identical laterality | Not reported | T-116 test | T-116 test | UC-HELP shooters were more accurate | UC-HELP | |
| *Pointer (2008)* | Motorsports | Crossed, Uncrossed laterality | RA (31.7, 68.3) NA (30, 70) | Author self-report questionnaire | Hole-in-the-card Test | No relation found | – | No effects found |
| *Portal & Romano (1988)* | Baseball | Crossed, Uncrossed laterality | RA (35, 39) NA (18, 65) | Direct preference observation | Pointing Test | There are twice C-HELP in the group of baseball players than in normal controls. | – | C-HELP |
| *Quevedo et al. (2014)* | Multi-Sport | Crosslateral, Homolateral dominance | HPA (39.9, 61.1) | Interview | Pointing Test and Sighting Test | There are more UC-HELP shooters and C-HELP in golf and team sports than in normal population | – | Golf and team sports: C-HELP Shooting: UC-HELP |

| Study | Sport | HELP terminology | C-HELP%, UC-HELP% | Handedness assessment | Eye preference assessment | HELP and sports performance/ skill level relationship | Favourable direct effects | Favourable indirect effects |
|---|---|---|---|---|---|---|---|---|
| *Razeghi (2012)* | Darts | Crosslateral, Unitaleral dominance | Not reported | Edinburgh Inventory | Hole-in-the-card and Pointing Test | No significant differences between C-HELP and UC-HELP in skill with darts | – | No effects found |
| *Shick (1971)* | Basketball | Contralateral, Unilateral dominance | Not reported | Direct preference observation | Hole-in-the-card test | UC-HELP registered more lateral errors towards de side of nondominant hand | C-HELP | – |
| *Shick (1977)* | Basketball | Contralateral, Unilateral dominance | RA (32.7, 67.2) | Direct preference observation | Hole-in-the-card test | No relation found on lateral errors in free-throw shooting for college women and HEL | – | No effects found |
| *Sugiyama & Lee (2005)* | Golf | Crossed dextral, Pure dextral | Not reported | Hand Dominance Questionnaire | Pointing Test | . | No effects found | |
| *Dallas, Mavvidis & Ziagkas (2018)* | Tennis | Contralateral, Ipsilateral dominance | HPA (42, 58) | Direct preference observation | Direct preference observation through pictures | There are more C-HELP in the 50 best world tennis players than in normal populations | – | C-HELP |
| *Zouhal et al. (2018)* | Soccer | Crossed, Non crossed laterality | HPA (53, 47) RA (33, 67) | Individual laterality in sports. *Azemar (2003)* | Individual laterality in sports. *Azemar (2003)* | There are more C-HELP in the soccer elite players than in the regular group. | C-HELP | |

**Note:**
C-HELP, hand-eye laterality crossed profile; UC-HELP, hand-eye laterality uncrossed profile; HPA, high-performance athletes; RA, regular athletes; BA, beginner athletes; NA, non-athletes; –, not assessed.

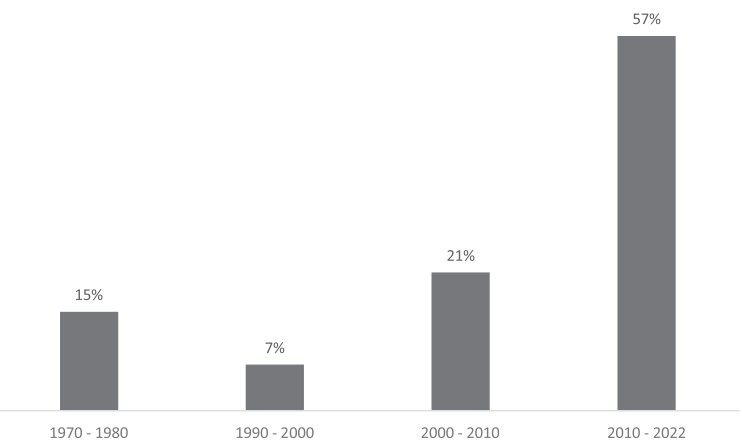

**Figure 2 Percentage of reviewed studies by publication date.**

the studies the relationship between hand-eye laterality and performance was studied in tennis, baseball, soccer, cricket, archery, biathlon, motorsports and darts, with one article for each discipline.

## Hand-eye laterality assessment

In laterality research, a wide range of assessment methods are continuously altered and developed. As it is a multidimensional phenomenon, many different tools try to measure the underlying variables. In the studies selected for this review, diverse and varied strategies for the evaluation of eyedness and handedness have been identified.

The following assessment types have been proposed by *Faurie, Raymond & Uomini (2016)* to better classify and identify the predominant methods used in the current literature:

1. Performance tasks: activities designed to induce actions from which a degree or level of laterality can be deduced.
2. Preference tasks: activities designed to induce direct spontaneous actions of a preferred side of the body.
3. Self-report questionnaire: questionnaires where the subjects decide whether they prefer one side or the other for different contexts and actions.
4. Other author assessment measures, including interviews and active observation by evaluators.

The measurement methods used in the selected studies have been classified in the next two subsections according to the variable they measure and the type of evaluation (Table 3).

### Handedness assessment

Handedness measurement is further divided into measures of preference and performance. While hand preference identifies the preferred hand for completing a task, hand performance differentiates between the ability or proficiency of one hand over the other in a particular task (*MacManus & Bryden, 1992*). There is debate over whether performance and preference measures are indicators of common underlying factors, or separate dimensions of behavior with different causes (*Bishop, 1989*).

Eight different methods have been used to identify hand preference. 62% of the studies used self-reported questionnaires, approximately 23% of the studies used direct observation as a method to determine preference, and 15% of the studies used other methods including performance tasks and interviews.

### Questionnaires

Three studies included a self-reported author questionnaire as an instrument to assess the handedness (*Dalton, Guillon & Naroo, 2015*; *Pointer, 2008*; *Quevedo et al., 2014*).

*Zouhal et al. (2018)* also used an author questionnaire validated with a sample of 1,500 athletes (*Azemar, 2003*). In this case the researcher filled out the questionnaire based on the direct observation of 11 performance tasks.

### Eyedness assessment

Different tests have been described to measure eyedness, and there is controversy in determining whether commonly used tests report an accurate evaluation of this

**Table 3 Handedness and eyedness assessment methods.**

| Assessment | Type | Instrument | Administration (items) | Author | Reliability | Studies[1] | Sport |
|---|---|---|---|---|---|---|---|
| Handedness | Test | Edimburgh Inventory | Self-reported (20 items) | Oldfield (1971) | Yes | 3 | Archery; Cricket; Darts |
| | | Hand Dominance Questionnaire | Self-reported (13 items) | Chapman & Chapman (1987) | Yes | 1 | Golf |
| | | Harris Test | Performance task (11 items) | Harris (1947) | Yes | 1 | Basketball |
| | Autor questionnaire | | Self-reported (1 item) | Dalton, Guillon & Naroo (2015) | – | 1 | Golf |
| | | | Self-reported (–) | Pointer (2008) | – | 1 | Motorsport |
| | | | Self-reported (3 items) | Quevedo et al. (2014) | – | 1 | Multisports |
| | Direct observation | | Observation of hand preference on the task of basketball | Shick (1971, 1977) | | 2 | Basketball |
| Eyedness | Direct observation | Pointing/Porta Test | Performance task | Porta (1593)[2] | Yes | 6 | Archery; Baseball; Darts; Golf |
| | | Hole-in-the-card-test | Performance task | Crider (1944), Coren & Kaplan (1973), Rice et al. (2008) | Yes | 4 | Basketball; Darts; Motorsport |
| | | Sighting/Miles Test | Performance task | Zazzo (1960)[2] | – | 2 | Basketball; Multisports |
| Handedness and eyedness | Test | T-116 Test | Performance task (12 items) | Matějček (2007) | – | 1 | Biathlon |
| | | Individual laterality in sports | Self-reported (11 items) | Azemar (2003) | – | 1 | Soccer |
| | Direct observation | | Web photographies | Dallas, Mavvidis & Ziagkas (2018) | | 1 | Tennis |

**Notes:**
[1] Number of reviewed studies applying the instrument.
[2] Most cited referring the instrument.
–, not reported.

phenomenon (*Laby & Kirschen, 2011*). Subsequently, we have described the different methods used and the variations incorporated by the authors in the selected studies.

*Pointing test or porta test*

The pointing test is the most frequently used procedure; six of the selected studies in this review applied this method. Is also known as the Porta test because the earliest known reference dates back to *Porta (1593)*. The pointing test tries to create a situation in which the two eyes cannot be used simultaneously. The subjects must align three points: the dominant eye, the finger and a distant target. The test starts by keeping both eyes open and proceeds by closing one eye at a time, which reveals the dominant eye (the eye that is aligned with the finger). Variations and modified versions have been found in the

**Table 4 Variations of the Pointing Test (Porta Test) depending on the target (type and distance), the pointing technique and the identification method of the dominant eye.**

| Study | Sport | Target | Target distance | Pointing technique | Assessment variations |
|---|---|---|---|---|---|
| *Dalton, Guillon & Naroo (2015)* | Golf | Chart (Michel Guillon vision clinic) | Scalable at any distance | Index finger on both arms alternately | Evaluators cover both eyes alternately, and subject indicate where the finger and target still aligned (that is the dominant eye) |
| *Laborde et al. (2009)* | Archery | Any object | >2 m | Index finger on one arm | Subject close one eye at a time. The eye aligned with the object and the finger is dominant sighting eye |
| *Mann, Runswick & Allen (2016)* | Criquet | Camera | 3 m | Thumb finger on both arms alternately in specific batting stance | Photograph |
| *Portal & Romano (1988)* | Baseball | – | – | – | – |
| *Razeghi (2012)* | Darts | Any object | – | One arm index finger | Subjects close the eyes alternately or draw the finger back to the head |
| *Sugiyama & Lee (2005)* | Golf | Examiner nose | – | Index or thumb finger on both arms alternately | The eye with which the finger was aligned was noted |

**Note:**
–, not reported.

application of the target (type and distance), the pointing technique and the identification method of the dominant eye (Table 4).

Regarding the variations in the target, two studies indicated that any object could be used as a target, and while *Razeghi (2012)* did not indicate the distance between the target and the subject, *Laborde et al. (2009)*, following *Buxton & Crosland (1937)* protocol, specified a minimum distance of 2 m between target and subject. *Dalton, Guillon & Naroo (2015)* used a specific chart developed at the Michel Guillon vision clinic, which is scalable at any distance. *Sugiyama & Lee (2005)* used the examiner's nose as a target with no indications of the distance. Finally, *Mann, Runswick & Allen (2016)* implemented the pointing test using a camera as a target at a distance of 3 m.

Regarding variations in the pointing technique, three of the studies (*Sugiyama & Lee, 2005*; *Dalton, Guillon & Naroo, 2015*; *Mann, Runswick & Allen, 2016*) used a finger (index or thumb) on both arms alternately to reduce the interference with handedness; this procedure was reported by *Porac & Coren (1976)*. The studies from *Razeghi (2012)* and *Laborde et al. (2009)*, both studying precision sports (darts and archery), used the index finger of a single hand to point; this single-handed procedure was validated by *Lora, Heilman & Roth (2002)*.

Finally, regarding the variations in the identification of the dominant eye, three different procedures were found: in two studies (*Laborde et al., 2009*; *Razeghi, 2012*) the subject actively closed each eye to determine which eye was aligned with the target. *Dalton, Guillon & Naroo (2015)* used a passive measurement to identify the dominant eye in which the examiner covered one eye of the participant, followed by the other eye, and asked participants to report the resulting deflection from the center of the target. In two studies,

the examiners observed which eye was dominant, using a photograph aligning the finger/thumb with the focus of the camera and observing in the photograph which eye was aligned (dominant) and which was covered by the hand (non-dominant) (*Mann, Runswick & Allen, 2016*). In one study they used direct observation while sighting (*Sugiyama & Lee, 2005*).

*Sighting test*

The sighting test, also known as the Miles Test, was initially introduced by *Zazzo (1960)*. According to *Laby & Kirschen (2011)*, this is one of the most common and easy behavioral tests to determine eye dominance. The procedure responds to a similar mechanism as the pointing test, aligning an object with a reference from our hands. In this test, instead of using a finger or a pen, the subjects are asked to hold their hands together, with their palms facing away at arm's length, in such a way that a small space remains between the thumbs and fingers of the two hands. We found two different procedures to determine the dominant eye: passive or active measurement.

For the passive measurement, the examiner covered one eye of the subject, followed by the other eye, and asked with which eye the target was no longer seen (*Dorochenko, 2009*; *Laby & Kirschen, 2011*).

For the active measurement, the subject brings their hands to their face quickly keeping the object in view, moving the hole in the hands to their dominant eye, thus indicating the dominant eye (*Knudson & Kluka, 1997*). *Quevedo et al. (2014)* used a sighting test without giving any further procedural information. Finally, *Lopez-Diaz et al. (2015)* applied both active and passive procedures.

*Hole-in-the-card test*

The hole-in-the-card test is also a behavioural-preference-sighting test. In this case, the subject holds a card with a hole in the middle at arm's length, he is instructed to focus both eyes on an object through the hole, then without taking his focus off he will bring the card closer to his face, directing the hole at the dominant eye (*Crider, 1944*; *Coren & Kaplan, 1973*). This measurement has been applied in four studies (*Shick, 1971*, *1977*; *Razeghi, 2012*; *Pointer, 2008*). The involvement of both hands in the card test and the sighting test allows handedness interference to be avoided.

*Observation of pictures*

One of the studies *Dallas, Mavvidis & Ziagkas (2018)* used observation of photographs of the athletes playing found on the web to evaluate the hand and eye dominance. This is a non-validated method that is based on a subjective assessment by the examiner.

## Terminology

Terminological dispersion has been observed among the studies when classifying hand-eye laterality profiles. The terms "crossed" *vs* "uncrossed", also seen as "crossed" *vs* "noncrossed" were the most common and were used in five studies (35.71%). The terms "crosslateral" *vs* "unilateral" or *vs* "homolateral" were used in two studies (14.29%). The terms "contralateral" *vs* "unilateral" were used in two studies (14.29%), both

by the same author. Other terms used were "crossed" *vs* "identical", "crossed" *vs* "homogeneous", "crossed" *vs* "pure", and "contralateral" *vs* "ipsilateral", each of them in a single article.

We have also found very diverse terminology regarding the categorization of the different skill levels. In the present study we have unified the terms and categorized four different groups: (a) a high-performance athlete (HPA) group, where we included all samples related with professional athletes who train full-time, like the best 50 tennis players in the world, the first division of soccer (league one in France), elite multisport athletes awarded with national grants at the national high performance center in Spain and golf players from the European Tour and Ryder Cup level; (b) a regular athlete (RA) group, where we included part-time athletes who, although they compete and train in a systematic way, are not professionals; in this group, we included amateur soccer players, Challenge Tour golfers (one step below the EuropeanTour), college students and junior level athletes; (c) a beginner athlete (BA) group, which included subjects with an elementary skill level, which could be considered a control population, but which were considered in the studies in relation to a specific sports skill; and (d) a non-athlete (NA) group, where we included random subjects without any relation to the sport studied.

## Distribution of laterality profiles

Assuming the distribution of 10–30% for C-HELPs in the general population reported by *Robinson, Jacobsen & Heintz (1997)*, or the 34.8% reported by *Bourassa, MacManus & Bryden (1996)*, we found a significant C-HELP overrepresentation in high-performance athletes for four different modalities: golf (52.55%), soccer (53%), tennis (42%) and team sports (50.7%) (Table 2; Fig. 3). However, the results also show a UC-HELP overrepresentation for some target sports: high-performance shooters (93,1%) and regular archers (82.3%). In that sense, *Erickson (2007)* noticed that in aiming sports such as target shooting or archery, the UC-HELP offers advantages in acquiring the skills required for success due to the specific homolateral demands of this sport (riffle and eye must be aligned on the same side to aim properly.

For the regular athlete sample, the only remarkable result is the overrepresentation of C-HELPs in the sport of golf. This was observed among a sample of golfers from the Challenge Tour, one step below the European Tour, who are still dedicated athletes with an advanced skill level (*Dalton, Guillon & Naroo, 2015*).

The results for the RA, BA and NA samples for the other sports are coincident with the distribution reported for the general population. One study by *Portal & Romano (1988)* indicated central ocular dominance (cyclopean eye) in baseball players, where the athletes eye preference is balanced, using a modified version of the pointing test. Four of the selected studies do not show information on laterality distributions, as they directly study differences in performance between hand-eye laterality profiles using different indicators.

## Effects on performance

The last columns of Table 2 also classify the results of the studies as follows to assess the effects on performance of both hand-eye laterality profiles: (1) direct effects, when

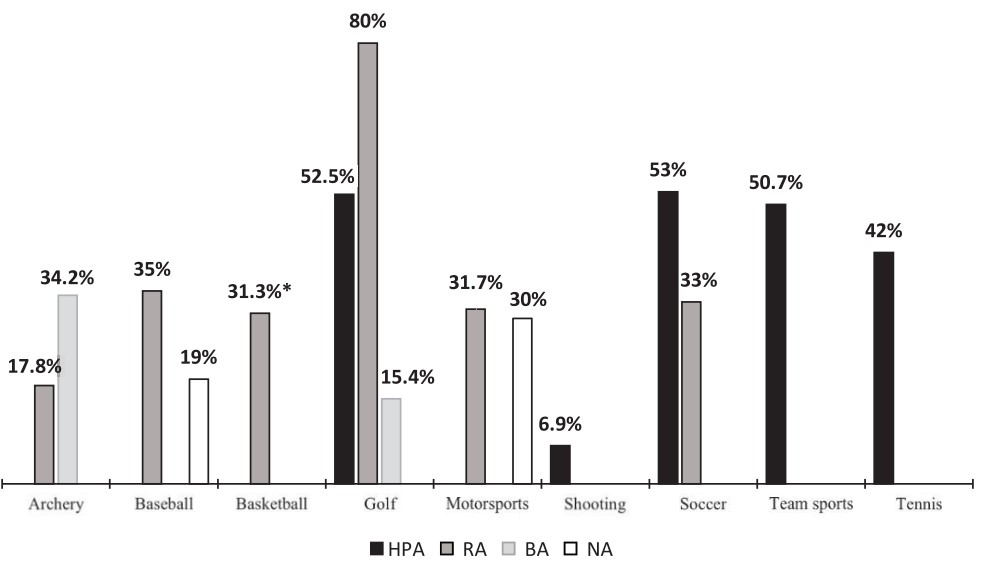

**Figure 3 Distribution of crossed profiles by sport modality and skill level.** HPA: high-performance athletes; RA, regular athletes; BA, beginner athletes; NA, non-athletes; *, weighted percentage.

performance indicators have been assessed; and (2) indirect effects, when a relative advantage is observed because of the over-representation of one profile over the other in the most skilled athletes.

Six different studies confirm performance enhancements of C-HELPs over UC-HELPs, including both direct and indirect effects. The results for baseball (*Portal & Romano, 1988*) showed an increase in the UC-HELP prevalence for the group of non-athletes (65%), in relation to the group of regular athletes (39%). We observed an indirect effect on golf performance in two different studies (*Quevedo et al., 2014*; *Dalton, Guillon & Naroo, 2015*) due to the enhanced distributions of C-HELPs in the HPA sample (55.1% and 50% respectively for the two studies). We also observed an increased percentage of C-HELPs (42%) in the top 50 tennis players in the world (*Dallas, Mavvidis & Ziagkas, 2018*). We consider an indirect effect on performance for soccer players because 53% of the HPA are C-HELPs (*Zouhal et al., 2018*). The results for basketball, however, are inconsistent; while *Shick (1971)* found a favorable direct effect that was later refuted (*Shick, 1977*), *Lopez-Diaz et al. (2015)* reported some distributions in regular athletes that are congruent with those of the normal population. Even though over-representations of C-HELPs are observed in the highest-level athletes, no study has shown direct effects on performance for the crossed profiles.

Three different studies confirm advantages of UC-HELPs in target sports (archery, biathlon and shooting) over C-HELPs, including both direct and indirect evidence. Archery and shooting are the only samples that show an overrepresentation of UC-HELPs in relation to the distributions in the normal population.

Four studies have found no relevant effects on performance related to hand-eye laterality profiles. *Shick (1977)* refuted the relationship found above in basketball players

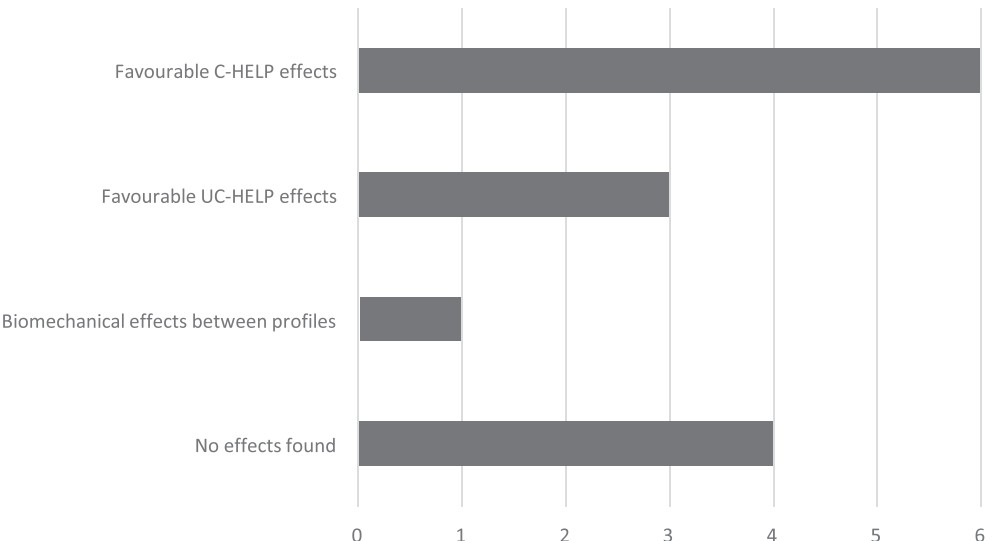

**Figure 4  Effects of laterality profiles reported by number of selected studies.**

(*Shick, 1971*) between UC-HELPs and lateral throwing errors, where UC-HELPs seemed to make more mistakes than C-HELPs. *Razeghi (2012)* did not observe differences in accuracy between darts players, conflicting with other reported results on the advantage of UC-HELPs in precision or target sports (*Erickson, 2007*). In motorsports (*Pointer, 2008*), we observed congruent distribution of laterality profiles between athletes and the normal population. *Sugiyama & Lee (2005)* concluded that more research is needed to confirm possible effects of hand-eye laterality profiles on golf putting stance.

Finally, one study shows biomechanical differences concerning hand-eye laterality profiles and a specific technique, which were unable to be categorized as positive or negative. The results on *Lopez-Diaz et al. (2015)* support that an alternative basketball shot technique for UC-HELP (rotating their body position 45°) will help them to obtain higher shot percentages.

Figure 4 summarizes the findings as to whether there is a favorable effect on performance for C-HELP over UC-HELP, for UC-HELP over C-HELP, if there is a biomechanical effect reported or if no effect is found. We observed reports of 11 performance effects that were related to hand-eye laterality profiles in selected studies, while four studies didn't report any effect.

### Psychological traits and hand-eye laterality

None of the reviewed studies provided data on the relationship between psychological traits and hand-eye laterality.

## DISCUSSION

The aim of this study was to systematically review the scientific publications on hand-eye laterality in sports, to estimate the prevalence of C-HELPs and UC-HELPs in different sports modalities and to examine their association with sports performance and

psychological traits. We would like to lay the groundwork for future research into the study of hand-eye laterality profiles in sports, considering the growing number of publications about this topic.

## Distribution of laterality profiles and effects on performance

The results referring to the distribution of the hand-eye laterality profiles according to level of practice, as well as the direct and indirect results found on performance, indicate that hand-eye laterality profiles could be considered as a valid performance indicator.

We used the figure from *Bourassa, MacManus & Bryden (1996)*, who found that 34.8% of the general population exhibited a C-HELP, as a control value to compare against sporting profiles, since they obtained a larger sample in their meta-analysis. In the studies included in our review, we observed that certain sports have different incidences of hand-eye laterality profiles than the normal population depending on the level of practice. These results mostly refer to the distribution of hand-eye laterality profiles in different sports and levels of practice, but they do not allow us to conclude that there is a direct relationship between these profiles and sports performance. Even so, relevant patterns have been found in this regard, which we discuss below.

The C-HELP percentage reported for regular and high-level athletes of certain sports is higher than in the normal population, 52.5% in golf (*Dalton, Guillon & Naroo, 2015*; *Quevedo et al., 2014*), 42% in tennis *Dallas, Mavvidis & Ziagkas (2018)*, and between 50.7% and 53% in soccer, volleyball, handball, basketball, hockey, softball, and water polo (*Quevedo et al., 2014*; *Zouhal et al., 2018*). As these data indicate, C-HELP subjects seem to have performance advantages in these sports modalities. The explanation for the overrepresentation of C-HELPs in some sports seems to be complex. Some publications (*Siefer et al., 2003*) point to specific advantages for C-HELPs (especially those with left eyedness) in asymmetrical ball sports (tennis, soccer and basketball). Some literature focuses on the biomechanical effects of hand-eye laterality profiles, which modify and influence the specific movement, position, and technique of some asymmetric sports. For example, *Lopez-Diaz et al. (2015)* pointed out distinct technical adaptations in the basketball shot for the two profiles; while *Sugiyama & Lee (2005)* analyzed the differences in golf putting stances for the two profiles. There are also informative publications about tennis that reported accommodations of the hitting technique depending on the hand-eye laterality profile (*Garipuy & Wolff, 1999*). For instance, a right-handed player who predominantly perceived the ball with the right eye hit the forehand and served in a more frontal position than a right-handed player who was left eye dominant. This is because the sight from the dominant eye (perceptive input) and the racket on the dominant hand (motor output) must coincide at the point of impact of the ball and the player will naturally adjust his position based on both. For his part, *Dorochenko (2013)* also considered the advantage of the C-HELP for tennis performance in an informative, non-scientific publication. *Bache & Orellana (2014)* collected *Dorochenko (2013)* observations, pointing out that most of the Top 10 ATP tennis players are C-HELPs. We should also be cautious with the results for tennis given by *Dallas, Mavvidis & Ziagkas (2018)*, who reported an overrepresentation of C-HELPs (42%) in the world's top fifty tennis players, as we found

methodological inadequacies in eyedness assessment with indirect and non-standardized measurements (observation of images from the internet). Another hypothesis reported by *Azémar, Stein & Ripoll (2008)* tries to explain the advantage of the C-HELP in dual sports as the result of a shorter reaction time for C-HELP subjects.

In contrast, the C-HELP distribution recorded in target sports is extremely low, with 6.9% in high performance shooters (*Quevedo et al., 2014*), and 17.1% in regular archers (*Laborde et al., 2009*). Due to this data, UC-HELP subjects seem to have performance advantages in target sports modalities. The explanation for this phenomenon relies on a biomechanical argument, given that shooters and archers prefer to hold the weapon on the same side of the body as the dominant eye while aiming (*Jones et al., 1996*). In addition, the literature under review reported performance effects based on biomechanical differences in technical execution between the two profiles in some sports, such as basketball (*Lopez-Diaz et al., 2015*), cricket (*Mann, Runswick & Allen, 2016*), and golf (*Sugiyama & Lee, 2005*). In conclusion, it seems that laterality patterns may influence performance depending on the sport modality and that awareness of them could be a complement to talent detection and coaching development.

## Methodological and terminological consensus

In reference to the assessment of laterality, there is no homogeneity regarding the instruments used in the reviewed studies. Three different methods have been used to identify hand preference, 62% of the studies used self-reported questionnaires, approximately 23% of the studies relied on direct observation, and 15% of the studies used other methods including performance tasks and interviews. This lack of coherence stems from the different orientations of each study and each modality. We consider that to determine the handedness of asymmetric implement sports, such as tennis or fencing, direct observation is sufficient since the hand holding the racket or implement will reliably give us the hand preference information. Other asymmetric sports modalities that do not involve the grasp of an implement, such as basketball or soccer, or where the implement is wielded with two hands (golf, cricket, or baseball) may require more specific assessment types, like self-reported questionnaires or even performance tasks. On the other hand, the study of manual laterality in symmetric sports, such as cycling or swimming, would not have a special interest given the equivalent use of both body hemispheres.

To identify the dominant eye, four different methods have been used in the reviewed studies. The pointing test, the hole-in-the-card test and the sighting test are the most widespread protocols and have been used in 92% of the selected studies. These methods are all preference, behavioral and sighting tests, based on the mechanism of aligning an object with a reference from one's hands. The pointing test involves aligning a target with one finger, or a pen held with only one hand, a fact that may cause handedness interference (*Porac & Coren, 1976*). The hole-in-the-card test avoids handedness interference by holding a card with two hands, and finally, the sighting test seems to be equally reliable and more practical since no material is needed as it is implemented with two hands (avoiding handedness interference). It is remarkable that only one of the studies (*Portal & Romano, 1988*) considered a type of neutral or central ocular dominance. This form of laterality

should be considered when the subject sees from the bridge of the nose like a cyclopean eye in passive measurement, or when the test is repeated and the subject brings their hands once to each eye inconsistently in active measurement. On that topic, *Laby et al. (1998)* concluded that although the one-handed pointing test does provide the possibility of detecting central dominance, it appears to be highly dependent on which hand is used for testing due to handedness interference.

After analyzing the results of this study, we consider that it would be necessary to establish a single and universal method for the measurement of hand-eye laterality that would avoid dispersion between methods. In our opinion, given the previous explanations, the most complete protocol would be made up of the combination of the assessment of handedness with direct observation (for asymmetric sports with implements like tennis, fencing, table tennis *etc.*) and the Edinburgh Handedness Inventory for other sports, and the application of the sighting test for ocular dominance, considering the active measurement protocol (bringing hands from arms length to the eyes), and using the examiner's nose or a camera lens at 3 m of distance as a target, and also considering its repetition for detecting possible central dominance cases.

From our review, we have noticed an important terminological dispersion between the studies when referring to hand-eye laterality profiles. In some works, we also found the term "dominance" instead of "laterality" to refer to the preference for one side of the body over the other. We haven't found enough evidence to use either of the two general terms. However, we have chosen for our review the most widely used form in the available studies for referring specifically to the type of dominance or laterality: uncrossed profile (UC-HELP) when the dominant eye and hand are on the same side of the body and crossed profile (C-HELP) when the dominance of the hand and the eye are on opposite sides.

## Limitations and future lines of research

Concerning laterality profile distribution, one of the biggest limitations that we find is that the distribution of C-HELPs and UC-HELPs in the normal population (non-athletes) is not yet clear, and therefore we cannot compare sports values with a standard value. While the hand-eye laterality meta-analysis of *Bourassa, MacManus & Bryden (1996)* compiled a 34.8% prevalence of C-HELPs, *MacManus et al. (1999)* reported a range between 24% and 27% for C-HELP prevalence, and other studies reported a range between 10% and 30% crossed (*Robinson, Jacobsen & Heintz, 1997*). Further investigation is needed to clarify and determine more objective and recent data about hand-eye laterality profiles in the general population.

It is clear that the study of laterality profiles is not as relevant in sports with "symmetrical" laterality, such as swimming, cycling or athletics (footraces), as it is in sports which require asymmetrical actions for throwing, hitting or shooting. In that sense, it would be necessary to corroborate the results and hypotheses about the effects of laterality profiles on performance in "asymmetrical" sports such as soccer, tennis, basketball, or hockey, and in target sports such as archery or shooting.

The methods or measurements used to establish the favorable C-HELP distribution in some studies are unknown and not published, as in the studies of *Dorochenko (2013)*

or *Bache & Orellana (2014)*, while in other studies they are improper and subjective, as in the work of *Dallas, Mavvidis & Ziagkas (2018)*. This could lead to hasty conclusions in sports like tennis, where more data is needed. In addition, to clarify these possible relationships between performance and hand-eye laterality, studies on specific performance indicators would be convenient, in addition to the standardization of the methods for assessing laterality.

Another goal we had set ourselves in this review was to relate the hand-eye laterality profile with psychological traits of the athletes. Some recognized experts from different disciplines point to a relationship between the different hand-eye laterality profiles and certain behavioral models, associating the dominance of the eye with the corresponding cerebral hemisphere (*Dorochenko, 2009*). Although this relationship and its applications seem to be very widespread in some specific areas such as professional tennis training, we did not find any study in our review that related laterality with psychological aspects of athletes. Research is needed on this possible association in the field of sports, through the application of behavioral and cognitive style questionnaires in conjunction with the application of consensual laterality tests, such as the Sport Orientation Questionnaire (SOQ) from *Gill & Deeter (1988)*, the Sport Competition Anxiety Test (SCAT) developed by *Martens et al. (1990)*, the Coaching Behavior Assessment System (CBAS) by *Smith, Smoll & Hunt (1977)*, the Revised Competitive Anxiety Inventory-2 (CSAI-2R) from *Cox, Martens & Russell (2003)*, or the Profile of Mood States (POMS) from *McNair, Lorr & Droppleman (1971)*.

## CONCLUSIONS

The study of the relationship between hand-eye laterality and sports performance is an underdeveloped field of knowledge, although it is notable that more than half of the publications found in this review are from the last decade. Our review provides information that could help shape future research in this area. Certain sports have different prevalences of hand-eye laterality profiles than the normal population. In sports such as golf, tennis, and team sports (soccer, volleyball, handball, basketball, hockey, softball, and water polo) the percentage of C-HELP is higher in regular and high-level athletes than in the normal population. In target sports (archery and shooting) the UC-HELP seems to confer an advantage, given the significant concentration of this profile in the highest performing populations, and some studies directly confirm these effects on biathlon shooting. In basketball, cricket and golf, the literature under review reported biomechanical differences between the two profiles in the execution of some techniques. It is worth highlighting the need for further scientific research on the distribution of hand-eye laterality profiles in asymmetrical sports like tennis, golf, basketball, or soccer, in order to study the mechanisms that produce direct effects on performance. The results shown in this review must be taken with caution as many of them refer to indirect effects.

We did not find any study in our review that related hand-eye laterality with psychological aspects of athletes. The incorporation of cognitive and behavioral indicators would provide very valuable information about the relationship between hand-eye laterality profiles and psychological or tactical sports patterns. In short, the advancement

of knowledge about hand-eye laterality could also contribute to more effective athlete development plans and could complement talent detection.

Finally, to ameliorate the terminological dispersion that we found in our review, we propose the term hand-eye laterality profile as a general topic and crossed profiles (C-HELP) and uncrossed profiles (UC-HELP) as the specific patterns. We also propose a combination of direct observation and the Edinburgh Handedness Inventory in handedness, and the application of the sighting test for eyedness as a protocol for hand-eye laterality measurement in sports.

### Funding
This study was supported by the grants PID2019-107473RB-C21 and PGC2018-100675-B-I00 from the "Ministerio de Ciencia e Innovación" of the Spanish Government.
The funders had no role in study design, data collection and analysis, decision to publish, or preparation of the manuscript.

### Grant Disclosures
The following grant information was disclosed by the authors:
Ministerio de Ciencia e Innovación of the Spanish Government: PID2019-107473RB-C21 and PGC2018-100675-B-I00.

### Competing Interests
The authors declare that they have no competing interests.

### Author Contributions
- Miquel Moreno conceived and designed the experiments, performed the experiments, analyzed the data, prepared figures and/or tables, and approved the final draft.
- Lluis Capdevila conceived and designed the experiments, analyzed the data, authored or reviewed drafts of the article, and approved the final draft.
- Josep-Maria Losilla conceived and designed the experiments, analyzed the data, authored or reviewed drafts of the article, and approved the final draft.

### Human Ethics
The following information was supplied relating to ethical approvals (*i.e.*, approving body and any reference numbers):
The Ethics Commission for Human Experimentation of The Universitat Autònoma de Barcelona(CEEAH-5745) granted Ethical approval to carry out the study.

### Data Availability
This is a systematic review with no raw data or code.

## Supplemental Information

Supplemental information for this article can be found online at http://dx.doi.org/10.7717/peerj.14385#supplemental-information.

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
