# Peer review of "Could hand-eye laterality profiles affect sport performance? A systematic review"

_PeerJ, doi:10.7717/peerj.14385_

## Round 0.1 · original submission · Major Revisions

As you will see, two referees are relatively enthusiastic about the paper. For one reviewer, there are a few more issues that deserved some of your attention. The comments are very constructive and clear.

·

Basic reporting

Clear and unambiguous, professional English used throughout.
Literature references, sufficient field background/context provided
Professional article structure, figures, tables. Raw data shared
Self-contained with relevant results to hypotheses.
All correct and interesting. I believe the paper meets the standards

Experimental design

Methods for the systematic review are properly described. Meet the standards.

Validity of the findings

Impact and novelty. Conclusions are well stated, linked to goals.

Additional comments

line 473 Quevedo (2015) is a mistake. Should be Quevedo, 2014

Reviewer 2 ·

Basic reporting

There is some very relevant literature missing, particularly on handedness in sport. The general literature on hand and eye dominance is well covered but the sport specific literature is not well addressed. Below is a recent edited book on laterality in sport that would be a good place to start. The editors such as Florian Loffing have published widely on laterailty in sport and barely feature here, so checking them out would be beneficial to building your rationale and conclusions.

https://books.google.co.uk/books/about/Laterality_in_Sports.html?id=MBOKCgAAQBAJ&printsec=frontcover&source=kp_read_button&hl=en&newbks=1&newbks_redir=1&redir_esc=y

Experimental design

I think the fundamental question here is 'is there enough evidence in this field to warrant a systematic review'?

It is true that this topic has not reviewed before, but that doesn’t offer strong justification for the need to do this work and the research questions do not reflect the state of the literature in this area. The review is conducted nicely but it feels like a premature time for a systematic review, very few papers have studied this question, particularly on the combination of eye and hand dominance. Those that have are very different and often answering very different questions.

Furthermore, work in binocular tasks in sport has found no effect of eye dominance on performance, so strong justification for the need to address that question is needed.

Prior reading should have quickly made it clear that no research has ever looked at this laterality profiles and psychological aspects in sport so again trying to conduct a systematic review addressing that question feels very strange and this element should be removed.

More work has come out since this search was done that would be relevant to the question (e.g., https://www.tandfonline.com/doi/full/10.1080/02640414.2021.1997011). But still something more like a scoping review would seem more appropriate, including questioning whether there is a need to look at ocular dominance at all outside monocular aiming tasks and aiming to develop a picture of the field and guide the primary work that is needed rather than expecting to make any strong conclusions on such a limited evidence base.

Validity of the findings

There are some issues to raise about interpretation of some of the papers here. For example, you mentioned that (line 484), 'Mann, Runswick & Allen (2016) proved how a specific cricket batting technique is more adaptative for C-HELPs.' This study did not measure batting technique and suggested there was no effect at all of ocular dominance and that handedness was solely responsible for any differences.

Additional comments

I think this can be a useful piece of work, but should not be presented as a systematic review when this area is in need of far more primary research to justify the questions this paper tried to address.

Reviewer 3 ·

Basic reporting

The manuscript is well-written and provides good background and rationale. Please see a few minor notes in the comments within the manuscript regarding grammar and figures.

Experimental design

No comment. The study was well designed, and I believe the question is relevant.

Validity of the findings

Findings are reported in a clear and concise manner. Conclusions, limitations, and future directions are included. Please see my comments in the manuscript. Perhaps most importantly, I feel that more information on biomechanical technique changes related to laterality is needed. If the data is available, it would be helpful to include information about the specific type of technique changes associated with crossed/uncrossed profiles in certain sports.

Additional comments

I commend the authors for taking care to be thorough in methods and reporting. Please see my full comments in the annotated PDF. Each highlighted text has an associated comment.

Annotated reviews are not available for download in order to protect the identity of reviewers who chose to remain anonymous.

---

## Round 0.2 · accepted · Accept

Accept in current form. Congratulations!

Reviewer 2 ·

Basic reporting

The authors have not offered changes but have given logical rebuttal to the previous comments.

Experimental design

As above, the authors have not made changes to the manuscript based on comments but have offered some constructive rebuttals.

Validity of the findings

Appropriate changes made